# OpenReview forum: "Signed Laplacians for Constrained Graph Clustering"
_ICML.cc/2025/Conference — ICML 2025 spotlightposter_

### Official Review · Reviewer_VAfm · 2025-03-12

**Overall Recommendation:** 3

**Summary:**

This paper addresses the constrained graph clustering problem, where the goal is to partition a graph into clusters while incorporating domain knowledge in the form of MUST-LINK and CANNOT-LINK constraints. The authors establish a Cheeger-type inequality that relates the solution of the constrained clustering problem to the spectral properties of the graphs $G$ and $H$. The proposed algorithm solves a generalized eigenvalue problem and demonstrates performance improvements over traditional spectral clustering methods.

**Claims And Evidence:**

The claims are well-supported.

**Essential References Not Discussed:**

n/a

**Experimental Designs Or Analyses:**

The experimental design is proper.

**Methods And Evaluation Criteria:**

This work addresses the problem of constrained clustering, where two graphs, $G$ and $H$, are provided as input. However, the experiments section solely compares the proposed method with the original spectral clustering algorithm, which is not inherently designed for constrained clustering tasks. Although Appendix B includes a comparison with the FC algorithm, it is important to note that FC represents an outdated baseline and does not reflect state-of-the-art performance.

**Other Comments Or Suggestions:**

Appendix B writes "This section will be further updated with more comparative methods as we expand our experiment"

**Other Strengths And Weaknesses:**

* The establishment of a Cheeger-type inequality provides a strong theoretical foundation, linking the constrained clustering problem to the spectral properties of the graphs.
* While the idea of adding self-loops to ensure the invertibility of the Laplacian matrix, thereby accelerating computation, cannot be claimed as entirely original, it demonstrates a notable level of creativity.

**Questions For Authors:**

1. Does the "sweep-set algorithm" in Algorithm 1 refer to kmeans?
2. Does the proposed method generalize to k-way (k > 2) clustering problems?

**Relation To Broader Scientific Literature:**

The established Cheeger-type inequality for constrained clustering is a generalization of the classical one as identified in Remark 3.3

**Theoretical Claims:**

I checked the proof sketch and it appears to be correct.

---

> ### Author Rebuttal · Authors · 2025-03-28
>
> We thank the reviewer for  their   positive evaluation and and valuable suggestions.  Here is our response to the raised questions:
>
> **Response to _Methods and Evaluation Criteria_:**
>
> We agree that further comparison between our work with the state-of-the-art constrained clustering algorithms will significantly improve the value of our work. Following the suggestion, we will add more experimental results in the next version of our paper.
>
> **Response to Q1:**
>
> > Does the "sweep-set algorithm" in Algorithm 1 refer to $k$-means?
>
> No. The sweep-set algorithm used in Algorithm 1 is a spectral procedure commonly used in classical Cheeger inequalities. This method involves sorting vertices according to their corresponding entries in the eigenvector solution obtained from the generalized eigenvalue problem. Then, the optimal clustering cut is identified by sequentially evaluating the constrained cut ratio for each prefix set of the sorted vertices.
>
> It is worth mentioning that, in our preliminary experiments, we tried to replace the sweep-set step with $k$-means on the eigenvectors, and received almost identical results as the current approach. Hence, we decided to retain the classical sweep-set approach in our final experiments to maintain consistency with the classical Cheeger inequality applications. We will add necessary discussion in the next version of the paper.
>
> **Response to Q2:**
>
> > Does the proposed method generalize to $k$-way ($k>2$) clustering problems?
>
> Generalizing our approach to the $k$-way  clustering  for $k>2$  is an important and natural direction of research, and we are currently working on this direction. Our initial work has indicated that  the techniques required for the multi-way extension differ substantially from our currently presented one. Specifically, it requires us to develop a new objective function and substantially different spectral arguments to analyze the algorithm's performance. Given the  amount of work needed for this generalization, we decided to make it as a separate and future work.
>
> Thank you once more for the suggestions and  questions. We're happy to answer any further questions during the discussion phase.

---

> > ### Comment · Reviewer_VAfm · 2025-04-03
> >
> > Thank you for the response. I have no further questions and have decided to maintain my score.

---

### Official Review · Reviewer_eq7q · 2025-03-14

**Overall Recommendation:** 3

**Summary:**

The paper provides a spectral method for approximately optimising the cut ratio for a graph $G$ and a constraint graph $H$ using the smallest non-zero eigenvalues of two particular graph Laplacians. This is based on a proof of Cheeger-type inequality similar to the min-cut approach for a single graph. Practical considerations are made to approximate this problem using a signed Laplacian which provides computational speed-up and improved results. Small experiments are run on two synthetic datasets and one temperature/spatial real-world example.

**Claims And Evidence:**

Main claims of paper where theoretical claims (see below).

It wasn't clear to me why the approaches in the related work at the top of page 2 were unsuitable for this problem. Why would sweep-cut versions based on these inequalities not work for the constraint clustering problem? Below this, the authors mention papers with a practical perspective (lines 068-070), which are not compared against in the experimentation, which may still be useful if currently lacking the same theoretical rigour.

**Essential References Not Discussed:**

N/A

**Experimental Designs Or Analyses:**

Real-world example could be more impactful given some of the image segmentation experiments explored in referenced CC papers.

The metric used for successful separation in the temperature experiment is unusual. It would be more natural to perform a statistical test to see if the temperature are significantly different to better account for more variance when one cluster has relatively fewer data points.

**Methods And Evaluation Criteria:**

It seemed unfair to compare CC and CC++ using graphs $G$ and $H$ only to spectral clustering techniques that only considered $G$. If the metric is to optimise ARI, then spectral methods acting jointly on $G$ and $H$ (or $G$ and $\bar{H}$) could be use, for example, using the unfolded adjacency matrix or its Laplacian. I think a fairer and wider comparison necessary for the paper to be accepted.

**Other Comments Or Suggestions:**

"Cheer-type inequalities" should be "Cheeger-type inequalities" on page 1 by Related work.

**Other Strengths And Weaknesses:**

Interesting problem that I have not considered before and I very much enjoyed learning about the theory of spectral CC. More motivation for the real-world application would help explain why this is a relevant problem as I felt I didn't get that feeling under the very end of the paper after reading some of the referenced work.

**Questions For Authors:**

Is it possible to create a corollary of Theorem 3.2 that expresses the upper bound in terms of the original graphs $G$ and $H$ rather the degree sequence equalised versions?

**Relation To Broader Scientific Literature:**

I do not know how much work relies on solving the graph constraint clustering problem.

**Theoretical Claims:**

Theorem 3.2 gives an upper bound for $\Phi_G^H$ with a convincing proof and sketch proof. I need more convincing about the statement about the approximation at the bottom of page 5 especially as the paper does not outline how the negative self-loop weight should be chosen.

---

> ### Author Rebuttal · Authors · 2025-03-30
>
> We  thank the reviewer  for their thoughtful review and valuable suggestions. Here is our response to the raised questions.
>
> **Response to _Claims And Evidence_:**
>
> >  Why would sweep-cut versions based on these inequalities not work for the constraint clustering problem?
>
> In our setting, sweep-cut is adapted from the classical Cheeger setting to work with eigenvectors of the generalized eigenproblem $ \Delta^G x = \lambda \Delta^H x $. Although standard in spectral clustering, here the procedure ensures that cuts reflect the joint structure of both graphs. We will clarify this briefly in the next version of the paper.
>
> **Response to _Methods And Evaluation Criteria_:**
>
> We agree that further comparison with state-of-the-art constrained clustering methods would enrich the paper. In this version, we compared our method with classical spectral clustering (SC), flexible constrained spectral clustering (FC), and our variants using self-loops (CC++). These are meaningful baselines given our focus on spectral guarantees. However, we acknowledge the importance of broader comparisons and plan to include these in the next version of our paper.
>
> **Response to _Theoretical Claims_:**
>
> The self-loop weights method serve a dual purpose. First, adding self-loops to $G$ balances the graph by equalizing the degree sequences of $G$ and $H$ (as noted in the manuscript, lines 118–119, right column). Second, including a small self-loop (with weight $\epsilon = 0.0001$) in $H$ ensures that the Laplacian $\Delta^H$ is invertible, which is necessary for solving the generalized eigenvalue problem $\Delta^G \mathbf{x} = \lambda\Delta^H \mathbf{x}$. This perturbation $\epsilon$ provides numerical stability, and we observed that the performance remains robust for other comparably small values of $\epsilon$.
>
> **Response to _Experimental Designs Or Analyses_:**
>
> We appreciate the suggestion to improve the statistical evaluation of the temperature dataset. In our current version, we evaluated the separation via differences in mean temperature between clusters. We agree that formal statistical tests (e.g., t-tests) would provide stronger evidence and intend to include them in a future revision. Despite this, our current metric demonstrated effective alignment between spatial structure and temperature variation. Nonetheless, even the simple metric showed that the separation achieved by our method aligned well with the underlying temperature variation.
>
> **Response to _Other Comments Or Suggestions_:**
>
> Thank you for catching the typo (“Cheer-type”). We will correct this in the next version of the paper.
>
> **Response to _Questions For Authors_:**
>
> We thank the reviewer for this insightful question. Indeed, we are currently exploring an extension of Theorem 3.2 that avoids the degree-sequence equalization and instead expresses the bound using the original graphs $G$ and $H$. To do this, we consider the more general setting of weighted graphs with vertex weights.
>
> Specifically, for weighted graphs $(G, w^G)$ and $(H, w^H)$, we assign weights to the vertices such that $w^G(v) = w^H(v)$. While the normalized Laplacian typically assumes $w^G(v) = \deg_G(v)$, our setting leads to a weighted unnormalized Laplacian $\Delta^{(G, w)}$. This operator remains positive and self-adjoint, and satisfies that $\lambda_2(\Delta^{(G, w)}) > 0$.
>
> In this formulation, the degree-sequence equalization step becomes unnecessary, albeit at the cost of working with a more general Laplacian operator. We believe a similar Cheeger-type inequality can be established under this setting, though doing so will require both theoretical and algorithmic adjustments. This is an interesting and nontrivial direction for future research rather than a mere corollary of the current work.

---

### Official Review · Reviewer_SYBT · 2025-03-16

**Overall Recommendation:** 4

**Summary:**

The paper considers the constrained graph clustering problem.
The input consists of two graphs, G and H, defined on the same set of nodes.
The clustering challenge is to group together nodes that are connected with large
weights  in G and small weights in H.

The paper considers only clustering into two clusters, which requires
identifying a single subset S of nodes. It defines a clustering criterion
as the ratio between the cut of S in G and the cut of S in H.
Clustering is achieved by finding S that minimizes this criterion.

The paper proves an upper bound on this criterion, which can be expressed
in terms of eigenvalues. The proof is constructive, and suggests a method
of computing S that achieves the bound. Still, the solution requires solving a
generalized eigenvalue problem. The paper shows how to do that efficiently
by introducing the signed Laplacian, a generalization of the standard Laplacian
operator.

The paper evaluates the algorithm on artificial and real data.
They show that the algorithm is practical, and compares favorably
with other clustering techniques (that operate only on G without a knowledge of
H).

**Claims And Evidence:**

Yes.

**Essential References Not Discussed:**

N/A

**Experimental Designs Or Analyses:**

Yes.

**Methods And Evaluation Criteria:**

Yes.

**Other Comments Or Suggestions:**

N/A

**Other Strengths And Weaknesses:**

As someone who is not familiar with this area I like this result very much.
The presentation is very clear,
the proofs are deep, and as much as I can tell they are valid.

My main concern is with identifying applications of this approach.
Where would the graph H come from?

Another point I am uncomfortable with is that the criterion does not
reduce to normalized cut when H is selected as the complete graph.
Instead, it reduced to regular cut, which is known not to be useful
in data analysis.

**Questions For Authors:**

N/A

**Relation To Broader Scientific Literature:**

I am not familiar with the constrained clustering. But the paper discusses
in detail comparison to recent results that are most likely the current
state of the art.

**Theoretical Claims:**

Only superficially.

---

> ### Author Rebuttal · Authors · 2025-03-30
>
> We thank the reviewer for their positive evaluation and insightful questions. Here is our response to the raised questions:
>
> > My main concern is with identifying applications of this approach. Where would the graph $H$ come from?
>
> We agree that clarifying the role and origin of the constraint graph $H$ is essential for broader applicability. In practice, $H$ encodes side information, domain knowledge, or external constraints — often available in real-world settings but not directly encoded in the similarity graph $G$. Examples include:
>
> - _Image segmentation:_ The graph $G$ connects each pixel to its spatial neighbors (e.g., via a 4- or 8-connected grid), with edge weights reflecting feature similarity such as color or texture. The constraint graph $H$ is constructed from user-provided annotations (e.g., brush strokes indicating foreground/background): MUST-LINK edges are added between pixels marked as belonging to the same object, and CANNOT-LINK edges are introduced between pixels annotated as different regions. This enforces spatial coherence guided by user intent.
>
> - _Social networks:_ In signed or trust networks, $G$ encodes positive relationships (e.g., friendship, following, co-authorship) where the presence of an edge implies mutual affinity. In contrast, $H$ captures negative interactions (e.g., distrust, blocking, or rivalry) by placing edges between users that should not be clustered together. This dual representation supports community detection that respects both cooperation and conflict.
>
> - _Complement graph:_ A practical and widely applicable construction is to set $H = \overline{G}$, i.e., the complement of $G$. Here, edges in $G$ represent strong (positive) similarities — interpreted as MUST-LINK constraints — while the absence of an edge in $G$ is interpreted as a weak or negative relation, thus becoming a CANNOT-LINK constraint in $H$. This approach is especially useful when no explicit constraint data is available, and could be defined as a default setting $H$ in that case.
>
>  We will clarify this construction and its practical relevance in the introduction of the revised manuscript.
>
> > Another point I am uncomfortable with is that the criterion does not reduce to normalized cut when $H$ is selected as the complete graph.
>
> Thank you for raising this thoughtful point. In fact, in earlier versions of our manuscript we included the following remark after Theorem 3.2, which answered the question above but was dropped due to page limit. We will add the the following remark in the next version of the paper.
>
> _Remark 1_: Theorem 3.2 can be viewed as a generalization of the classical Cheeger inequality. Specifically, if we consider the graph $H$ as the complete graph with $w_{uv} = 1$ for all edges, then it is straightforward to show that
> 	$$
> 	\min_{\emptyset \subset S \subset V} \frac{w_G(S, V \setminus S)}{|S| \cdot |V \setminus S|} \leq 4 \sqrt{ \lambda_2(\Delta^G)},
> 	$$
> 	where $\lambda_2(\Delta^G)$ is the second smallest eigenvalue of the normalized graph Laplacian of $G$.  	Similarly, if we consider the graph $H = (V, E', w^H)$ as the complete graph with self-loops where
> 	$$
> 	w^H_{uv} = \frac{\deg^G(u)\deg^G(v)}{\mathrm{vol}(G)},
> 	$$
> 	then
> 	$$
> 	\min_{\emptyset \subset S \subset V} \frac{w_G(S, V \setminus S)}{\min(\mathrm{vol}(S), \mathrm{vol}(V \setminus S))}
> 	\leq \min_{\emptyset \subset S \subset V} \frac{\mathrm{vol}(G)\, w_G(S, V \setminus S)}{\mathrm{vol}(S) \cdot \mathrm{vol}(V \setminus S)}
> 	\leq 4 \sqrt{ \lambda_2(\Delta^G)}.
> 	$$
> These reductions confirm that our constrained cut formulation includes both the regular (combinatorial) and normalized Cheeger cuts as special cases, depending on how the constraint graph $H$ is chosen.
>
> We thank the reviewer once more   for the suggestions and  questions. We're happy to answer any further questions during the discussion phase.

---

### Official Review · Reviewer_wvht · 2025-03-16

**Overall Recommendation:** 4

**Summary:**

This paper considers the constrained clustering problems over two graphs.
This paper establishes the Cheeger inequality for the proposed algorithm for constrained clustering, which can be a counterpart of Cheeger inequality to the standard spectral clustering over a graph. The proposed algorithm improves spectral clustering, in a scenario that is challenigin for spectral clustering. This paper also experimentally demonstrates the effectiveness of the proposed algorithm.

**Claims And Evidence:**

Claims seem sound. This paper provides the Cheeger type inequality for the constrained clustering problem. As far as written in related work, no previous work provided the proposed type of the Cheeger inequality. The proposed inequality is elegant; that is well adapted from the classical Cheeger inequality, and see clear contrast between the proposed and classical inequality.

Some may criticize this paper from a view where this paper only provides the bound when the number of clusters is two. However, in the classical Cheeger inequality for higher order has a different flavor even in the classical setting, like (Lee et al., 2014). Thus, I expect the higher order Cheeger inequality may be similar to the classical setting (much different than the current discussion), and therefore I do not think that this paper is not enough for this view point; this can be a distinct future work.

**Essential References Not Discussed:**

This does not affect to my overall evaluation, but it is nicer if the author can wrap up the existing work around graph constrained clustering, maybe in Appendix. Also, it would be nice to emphasize that the Cheeger inequality is a very established one by citing the classical literature such as [1] and [2].

[1] N. Alon. Eigenvalues and expanders. Combinatorica, 6(2):83–96, 1986.
[2] N. Alon and V. D. Milman. $\lambda_{1}$, isoperimetric inequalities for graphs, and superconcentrators. J. Comb.
Theory B, 38(1):73–88, 1985.

**Experimental Designs Or Analyses:**

I think the experimental design is fair.

**Methods And Evaluation Criteria:**

I think the methods make sense; that is streamlined with the existing view form the spectral clustering community.

**Other Comments Or Suggestions:**

N/A



--
post rebuttal comment: I am satisfied with the answers and therefore I increased my score from three to four.

**Other Strengths And Weaknesses:**

N/A

**Questions For Authors:**

It is nice if the authors can answer the questions in my responses.

**Relation To Broader Scientific Literature:**

I am curious that in which setting the proposed bound is tighter than the existing bound

$
\Phi^{G}_{H} \leq 16 \lambda_{2}(\Delta^{G}_{H})/\Phi(G)
$

by Koutis et al. (2023), since the both provide the bound for $\Phi^{G}_{H}$.

**Theoretical Claims:**

Although I have not checked every single detail of the proof, the claim seems be valid as far as I read.

---

> ### Author Rebuttal · Authors · 2025-03-30
>
> We thank the reviewer  their positive evaluation,  and constructive feedback. Here is our response to their questions:
>
> **Response to _Relation To Broader Scientific Literature_:**
>
> Our Cheeger-type inequality improves previous related results (including that of Koutis et al. (2023)). Our main theoretical result  states that
> 	$$
> 	\Phi_H^G \leq  4\sqrt{ \frac{\lambda_2(\Delta^G_H)}{\lambda_2(\Delta^H)}},
> 	$$
> 	where $\lambda_2(\Delta^H)$ is the Fiedler eigenvalue of the Laplacian of constraint graph $H$. In contrast to Koutis et al., our bound does not rely on an auxiliary demand graph $D_G$. Instead, it incorporates both graphs $G$ and $H$ directly through the generalized eigenvalue problem. Koutis et al.'s bound is based on a product of isoperimetric constants $\Phi_G$, $\Phi_H^G$, $\Phi^G_{D_G}$, where the influence of $H$ is less direct. As a result, their bound remains tied to the structure of $G$ and a third graph $D_G$, whereas ours explicitly improves with the connectivity of $H$. Even as additional constraints are added to $H$, modelled by a graph $H + e$, our bound becomes tighter due to the fact
> 		$
> 		\lambda_2(\Delta^H) \leq \lambda_2(\Delta^{H+e}),
> 		$
> 		which reflects the improved connectivity of $H$. Hence, our inequality rewards richer and more informative constraint graphs.
>
> As a concrete example, let's consider the case in which   $H$ is the complete bipartite graph $K_{n,m}$ with normalized weights. The spectrum of its normalized Laplacian is known to be $ \{ 0, 1 \text{ (with multiplicity } n + m - 2), 2 \}$, so $ \lambda_2(\Delta^H) = 1 $. In this case, our bound simplifies to $ \Phi_H^G \leq 4 \sqrt{\lambda_2(\Delta_H^G)} $, which depends directly on the spectral relationship between $G$ and $H$.
> 		This contrasts with the bound in Koutis et al. (2023), which involves an additional graph $D_G$ and the product $\Phi_H^G \cdot \Phi_{D_G}^G$, making the analysis more involved. Our formulation, in settings like $K_{n,m}$, yields a cleaner and more interpretable bound in terms of the original input graphs.
>
> **Response to _Essential References Not Discussed_:** We thank the reviewer for the suggestion. In the next version of the paper, we will give a more detailed discussion on existing work around graph constructed clustering, and in particular emphasise the work on classical Cheeger inequality including [1] and [2].

---

> > ### Comment · Reviewer_wvht · 2025-04-01
> >
> > Thank you very much for the response. I am satisfied with these and hence I will increase my score from three to four.

---

### Decision · Program_Chairs · 2025-05-01

**Decision:**

Accept (spotlight poster)

**Comment:**

This paper introduces a new spectral method for constrained clustering, supported by a Cheeger-type inequality. It seeks to minimize the ratio of cuts of two graphs (with shared set of nodes). The theory is solid and well-written. The algorithm uses a generalized eigenproblem and performs well on synthetic and real data.

Reviewers agreed that the theoretical part is strong and the proofs are correct. The main concern was the limited experimental comparison. The method was mostly compared to standard spectral clustering and one older baseline. Reviewers also asked for more explanation about how the second graph H is chosen in practice. The authors explained real-world cases where H comes from, showed that the method reduces to known cases when H is complete, and promised to add more experimental baselines. Overall, this is a strong theoretical paper with a clear contribution. The authors responded well to concerns, and I recommend accept the paper.